# Combined Genome-Wide Association Study and Haplotype Analysis Identifies Candidate Genes Affecting Growth Traits of Inner Mongolian Cashmere Goats

**DOI:** 10.3390/vetsci11090428

**Published:** 2024-09-12

**Authors:** Xiaofang Ao, Youjun Rong, Mingxuan Han, Xinle Wang, Qincheng Xia, Fangzheng Shang, Yan Liu, Qi Lv, Zhiying Wang, Rui Su, Yanjun Zhang, Ruijun Wang

**Affiliations:** 1College of Animal Science, Inner Mongolia Agricultural University, Hohhot 010018, China; aoxiaofang666@163.com (X.A.);; 2College of Vocational and Technical, Inner Mongolia Agricultural University, Baotou 014109, China; 3Key Laboratory of Mutton Sheep Genetics and Breeding, Ministry of Agriculture, Hohhot 010018, China; 4Key Laboratory of Goat and Sheep Genetics, Breeding and Reproduction in Inner Mongolia Autonomous Region, Hohhot 010018, China

**Keywords:** growth traits, genome-wide association analysis, haplotype analysis, Inner Mongolian cashmere goats

## Abstract

**Simple Summary:**

Inner Mongolian cashmere goats (IMCGs) are an excellent local breed that formed due to natural selection and artificial breeding over a long time, and they are a world-class breed used for both cashmere and meat consumption. Growth traits are the key indicators of cashmere goats’ growth, development, and health. Therefore, in this study, based on resequencing data (20×), the molecular markers and candidate genes affecting the growth traits of Inner Mongolian cashmere goats (Erlangshan type) were identified through a genome-wide association study combined with haplotype analysis.

**Abstract:**

In this study, genome-wide association analysis was performed on the growth traits (body height, body length, chest circumference, chest depth, chest width, tube circumference, and body weight) of Inner Mongolian cashmere goats (Erlangshan type) based on resequencing data. The population genetic parameters were estimated, haplotypes were constructed for the significant sites, and association analysis was conducted between the haplotypes and phenotypes. A total of two hundred and eighty-four SNPs and eight candidate genes were identified by genome-wide association analysis, gene annotation, and enrichment analysis. The phenotypes of 16 haplotype combinations were significantly different by haplotype analysis. Combined with the above results, the *TGFB2*, *BAG3*, *ZEB2*, *KCNJ12*, *MIF*, *MAP2K3*, *HACD3*, and *MEGF11* functional candidate genes and the haplotype combinations A2A2, C2C2, E2E2, F2F2, I2I2, J2J2, K2K2, N2N2, O2O2, P2P2, R1R1, T1T1, W1W1, X1X1, Y1Y1, and Z1Z1 affected the growth traits of the cashmere goats and could be used as molecular markers to improve the accuracy of early selection and the economic benefits of breeding.

## 1. Introduction

The goat, a highly adaptable ruminant, ranks among the earliest animals domesticated by humans. Variations in the natural environments and economic conditions across its distribution areas have led to its differentiation into fur, milk, and meat goats [1]. According to the survey statistics from the Sheep Annals of Chinese Livestock and Poultry Genetic Resources, China currently hosts 69 goat breeds; these include fifty-eight local breeds, eight breed varieties, and three introduced breeds, which are categorized based on their production purposes into meat, wool, hair, and milk types. The cashmere goat is a local, fine breed that has developed over an extended period through both natural selection and artificial breeding. Recognized as the world’s foremost breed for cashmere production, it is primarily found in China, Mongolia, Iran, and various other Asian countries. In China, the notable varieties of cashmere goat include Hexi, Liaoning, and Inner Mongolian cashmere goats. Among these, Inner Mongolian cashmere goats (IMCGs) can be further classified into three distinct types—Erlanglshan, Arbas, and Alashan—based on their geographical origin, each exhibiting strong meat production capabilities and a uniform distribution of fat. Growth traits serve as critical economic indicators in livestock meat production, as they provide insights into the carcass quality, the meat yield, feed efficiency, and disease resistance [2,3].

Genome-wide association studies (GWASs) represent a methodology employed to identify single-nucleotide polymorphism (SNP) loci across the entire genome. This approach utilizes inter-population SNP data, referencing a template genome, to uncover the markers or candidate genes that exhibit significant associations with specific target traits through comprehensive genotype and phenotype correlation analyses [4,5,6,7]. The advancement of high-throughput sequencing technology and the sequencing of livestock genomes have positioned GWASs as the primary method for identifying the candidate genes associated with significant economic traits in livestock and poultry. In recent years, the genetic variations in litter size [8,9], teat number [10,11,12], coat color [13], wool [14,15], fat deposition [16], and disease resistance traits [17] of different sheep and goat breeds have been studied by the GWAS analysis method. The related molecular markers and candidate genes have been identified, which provide important information and guidance for sheep breeding and genetic improvement. Significant loci found using genome-wide association analysis may be found in the non-coding or intergenic regions, and the linked loci may really be the actual loci linked to certain traits. Compared to SNP markers, haplotypes made from connected, significant loci are more effective at identifying the genes or loci linked to specific characteristics [18,19,20]. Utilizing haplotype analysis, numerous molecular markers and candidate genes related to growth, reproduction, immunity, and disease traits in livestock and poultry have been identified. Among these, *MSTN*, *IGF1*, *BMP2*, *HHEX*, *FUBP3*, and *METTL3* have been associated with growth traits [21,22,23,24,25]; *IL0RB*, *IL23A*, and *PRNP* have shown significant associations with immunity and disease traits [26,27,28]; and *ARL5A* and *CACNB4* have been linked to reproductive traits [29].

In this study, genome-wide association analysis and haplotype analysis were conducted on the growth traits (including body height, body length, chest circumference, chest depth, chest width, tube circumference, and body weight) of IMCGs (Erlangshan type) using whole-genome resequencing. The molecular markers and candidate genes associated with these growth traits were identified. These findings may provide a theoretical foundation for the marker-assisted selection and whole-genome selection of growth traits in IMCGs.

## 2. Materials and Methods

### 2.1. Phenotypic Statistics and Correlation Analysis

The data used in this study were obtained from the Erlangshan Ranch of the Inner Mongolia Northpeace Textile Co., Ltd., Hohhot, China (the Erlangshan National Breeding Farm of the Inner Mongolian cashmere goat; latitude 41°49′ N and longitude 108°56′ E). They adopted the group management of goats, with a total of 7 herds, and the males and females were kept in separate herds. The breeding environment used in this experiment complies with the China National Standard “Laboratory Animal Environment and Facilities” (GB14925-2010), which covers the requirements pertinent to a typical animal laboratory facility. The animals were fed and used in these experiments following the appropriate regulations for their care.

The study’s experimental data included 6653 body weights of 3883 individuals measured from 2020 to 2023, as well as 1608 body measurements (body height, body length, chest circumference, chest depth, chest width, tube circumference) of 268 individuals measured in 2023, along with corresponding genealogical records. The analysis of the correlation between the growth traits was conducted using R software (V3.6.0).

### 2.2. Genotyping

Ear tissue samples from 404 IMCGs were collected using ear amputation forceps, immediately placed in liquid nitrogen, transported to the laboratory, and stored at −80 °C. DNA extraction from these samples was performed using the phenol–chloroform method. The quality of the extracted DNA was assessed using a NanoDrop 2000 spectrophotometer and agarose gel electrophoresis. Qualified DNA samples were subsequently stored at −20 °C for future use. The qualified genomic DNA samples were then randomly fragmented into 350 bp lengths using a Covaris ultrasonic crusher. The entire library was prepared by a terminal. The whole library was prepared through terminal repair, poly(A) addition, splice sequencing, purification, and PCR amplification. Following library construction, preliminary quantification was performed using Qubit 2.0, and the effective concentration of the library was accurately quantified using qPCR to ensure its quality. Upon passing the quality assessment, the library was sequenced on the DNBSEQ-T7 platform at 20× coverage, utilizing the PE150 sequencing mode.

### 2.3. Quality Control and Population Stratification Assessment

Fastp software (V0.20.0) [30] was used to filter Raw reads data into Clean reads data, and a genome index was built for the reference genome. The filtered GCF_001704415.1 was mapped using Burrows-Wheeler-AliGner (BWA) software (V0.7.17) [31]; SAMtools software (V1.8-20) was used to convert the sam file after the comparison into a bam file and to sort the bam file. The MarkDuplicates program in Genome Analysis Toolkit (GATK) software (V3.8) [32] was used to remove duplicate data from the sorted bam files and obtain the final bam file. The final bam file was indexed, and the vcf file was obtained by using the HaplotypeCaller module in GATK software for SNP mutation detection.

Plink (V1.90) software was used to convert the vcf files into ped and map files and further control them. SNPs with a genotype detection rate (call rates) < 95%, a minimum allele frequency (MAF) < 5%, and Hardy Weinberg equilibrium (HWE) test SNPs with *p* value < 10^−6^ were excluded. Plink (V1.90) software was used to conduct population structure analysis (PCA) and establish a kinship matrix to analyze the kinship among cashmere goats, and R (V3.6.0) software was used for visualization.

### 2.4. Genome-Wide Association Analysis

Since the phenotypic data for body weight contained repeated records for the years 2020–2023, to disaggregate the effects of permanent environmental effects (covariance between different records of the same individual) on phenotypic observations, according to the average information restricted maximum likelihood (AI-REML) method, the breeding value of 3883 goat weight traits was estimated by using the single trait repetition force model in the DMUAI module of DMU software. The resulting estimated breeding value was added to the residual to obtain the corrected phenotype value, which was modeled as follows:y_c_ = Xb + Z_1_a + Z_2_p + e
where y_c_ is the vector that corrects the phenotypic value; b is the vector of the fixed effects, including the herd-measured year, age, and birth type; a is the vector of the additive genetic effects, p is the vector of the permanent environmental effects, and I is the identity matrix. X, Z_1_, and Z_2_ are the correlation matrices of b, a, and p, respectively; e is the residual vector.

The fastGWA-MLM model in GCTA (V1.94.0 beta) software was used to analyze the growth traits of IMCGs. The phenotypic data of morphological features (body height, body length, chest circumference, chest depth, chest width, and tube circumference traits) were only recorded once in 2023, so the phenotypic data were directly used for GWAS analysis, and the corrected phenotypic value (breeding value + residual) was used for GWAS analysis of the body weight traits. The model was as follows:y = Xsnpβsnp + Xcβc + Xgβg + e
where y is the phenotype vector; Xsnp is a genotype vector, and its effect is βsnp. Xc is the fixed effect correlation matrix and its corresponding coefficient is βc. Fixed effects affecting morphological features include age and herd. Xg is the random effect; its effect is βg and e is the residual vector.

The study set the genome-wide significance threshold at P = 1 × 10^−6^ [1]. The genomic inflation factor (λ) was determined using the slope from a linear regression of observed versus theoretical quantiles in R (V3.6.0). Significant SNPs were marked as threshold lines on Manhattan plots, which, along with QQ-plots, were created using the CMplot package in R (V3.6.0).

### 2.5. Gene Annotation and Enrichment Analysis

Subsequently, gene annotation was performed with the goat reference genome (ARS1, GCF_001704415.1) for 500 KB upstream and downstream of significantly related SNP sites using Bedtools software. Subsequently, gene ontology (GO) and the Kyoto Encyclopedia of Genes and Genomes (KEGG) were analyzed through the DAVID database to screen candidate genes related to growth traits.

### 2.6. Population Genetic Parameter Estimation

Data statistics and analysis were conducted based on the genome-wide significant SNPs obtained by GWAS. According to the calculation principles of allele frequency, genotype frequency, homozygosity, and heterozygosity, the population genetic parameters were calculated by the self-designed Excel program and tested to see whether they were in accordance with the Hardy–Weinberg equilibrium principle.

### 2.7. Association Analysis of Haplotype Combination and Growth Traits

The haplotype block was constructed using LD BlockShow (V1.40) software. Subsequently, LD analysis of the significantly associated SNPs was conducted using Haploview software. Finally, SAS (V9.2) software was utilized to perform association analysis of haplotypes in order to identify those significantly associated with growth traits and to search for candidate genes within this haplotype segment for further exploration of GWAS results.

## 3. Results

### 3.1. Phenotypic Statistics and Correlation Analysis

The results of the descriptive statistical analysis for the phenotypic data related to growth traits are presented in Figure 1. The coefficient of variation for body weight was relatively high at 19.15%, indicating a wide distribution and potential for improvement through selective breeding. In contrast, the coefficients of variation for body height, body length, chest circumference, chest depth, chest width, and tube circumference were all below 15%. Specifically, the coefficient of variation for chest width was 13.61%, while body height exhibited the lowest coefficient at 6.17%. This suggests a non-uniform population distribution, which is crucial for maintaining genetic diversity. Correlation analysis revealed a positive correlation among all traits, with a particularly strong positive correlation between chest circumference and chest width (see Figure 2). As illustrated in Appendix A, the phenotypic data for all traits followed a normal distribution, making it suitable for further analysis.

### 3.2. Genotyping

A total of 34,248,064 SNPs were identified by whole genome resequencing and further quality control was conducted. After genotype detection rate filtering, Hardy–Weinberg equilibrium filtering, minimum allele frequency filtering, and individual detection rate filtering (--geno 0.05, --maf 0.05, --hwe 1 × 10^−6^, --mind 0.1), a total of 17,234,359 SNPs were obtained for subsequent analysis. These loci were evenly distributed across 29 autosomal pairs of cashmere goats (Figure 3).

### 3.3. Genetic Relationship Analysis and PCA Analysis

Plink software was used to analyze the population structure based on the pairwise IBS distance. It can be observed from Figure 4a that these individuals gathered into a cluster. The stratification phenomenon exists in the test population, so the population stratification factor should be considered in the mixed model. In addition, based on the data after the quality control, the G matrix of the kinship relationships among the IMCGs was constructed (Figure 4b). The results of the G matrix construction show that 72.87% of the individuals were far from each other, and the coefficient of kinship was less than 0; 25.11% of the individuals were closely related, and the coefficient of kinship was between 0 and 0.1. For the remaining 2.02%, the individual relationship coefficient was greater than 0.1, indicating that the relationship was very close. These findings suggest an increased risk of inbreeding among these individuals, and it is advisable to avoid mating them in future breeding programs.

### 3.4. Genome-Wide Association Analysis

In this study, genome-wide association analysis and gene annotation were performed on the growth traits of cashmere goats based on genome-wide resequencing data. A total of 284 SNPs and 714 related candidate genes significantly associated with growth traits were detected, mainly located on chromosomes 8, 11, 16, 24, and 26 (Appendix A). One SNP (chr7_29685358) was associated with chest circumference and chest width, and there was a positive correlation between chest circumference and chest width. The Manhattan plot and Q-Q plot of growth traits are shown in Figure 5. The expected value in the Q-Q plot of each trait was consistent with the observed value; the λ value was 0.959–1.097; the model was reasonable and the loci were tilted upward, indicating that the effect of these loci was larger than that of random effect traits and further indicating that these SNPs were significantly correlated with growth traits.

Based on the results obtained from the gene annotation, the DAVID database was used to conduct GO and KEGG enrichment analysis of candidate genes; the results are shown in Figure 6. GO function annotation results show that these candidate genes were enriched in positive regulation of cell proliferation (GO:0008284), growth factor activity (GO:0008083), and negative regulation of hippo signaling (GO:0035331), ATP binding (GO:0005524), and *TGFB2* was enriched in the positive regulation of cell proliferation and growth factor activity. *MAP2K3* was enriched in the negative regulation of hippo signaling and ATP binding. *MIF* and *BMI1* were enriched in the positive regulation of B cell proliferation. KEGG enrichment analysis showed that candidate genes were closely related to cell aging and muscle and skeletal muscle formation, for example, in positive regulation of the B cell proliferation pathway (*TGFB2*, *MAP2K3*) and growth hormone synthesis, secretion, and action (*MAP2K3*, *GHRHR*, *IGFBP3*). Detailed information on significant related SNPs and candidate genes screened based on GWAS and enrichment analysis is shown in Table 1.

### 3.5. Population Genetic Parameter Estimation

The population genetic parameters of the SNPs related to growth traits excavated by GWAS were estimated (Appendix A), and it was found that three genotypes were detected in 253 of the 284 SNPs and two genotypes were detected in 31 SNPs. The predominant genotypes among the 163 single nucleotide polymorphisms (SNPs) were identified as wild types. All the dominant genotypes of 57 SNPs were heterozygous, and all the dominant genotypes of 64 SNPs were mutant. The heterozygosity (He) of chr1_G74254339C>T (CD) and chr16_G20533282G>A (CW) was the lowest (0.086). The heterozygosity of chr2_G1726639C>T and chr29_G20110970T>C (TC) loci was 0.500, indicating high genetic diversity. The effective alleles (Ne) of the Inner Mongolian cashmere goat population ranged from 1.094 to 1.999; the polymorphic information content of 171 SNP mutation sites was low polymorphism (0 < PIC < 0.25) and the polymorphic information content of 113 SNP sites was moderate polymorphism (0.25 < PIC < 0.50). The Chi-square test showed that 54 SNPs significantly deviated from the Hardy–Weinberg equilibrium state (HWE) (*p* < 0.05) and could not be used for subsequent analysis; 230 SNPs were consistent with the Hardy–Weinberg equilibrium state (HWE) (*p* > 0.05), indicating that these SNPs were less affected by selection pressure and mutations and that the selection intensity of the mutation SNP could be appropriately enhanced.

### 3.6. Association Analysis of Haplotype Combination and Growth Traits

LD analysis of SNPs (HWE-compliant) associated with growth traits was conducted using Haploview software. As illustrated in Appendix A, SNPs significantly correlated with the BH, BL, CC, CD, CW, TC, and BW of IMCGs, which comprised seven, two, three, eight, two, two, and three blocks (Appendix A, Appendix A). The association analysis between haplotypes and phenotypes was performed using SAS software, with the results presented in Appendix A and Figure 7. In the haplotype combinations constructed based on SNPs related to growth traits, the BH of haplotype A’s A2A2, haplotype B’s B2B1, haplotype C’s C2C2, haplotype E’s E2E2, haplotype F’s F2F2, and haplotype G’s G3G2 were significantly higher than those of other haplotype combinations (*p* < 0.05). The BL of the I2I2 haplotype combination of haplotype I was significantly better than that of other haplotype combinations. The CC of the J2J2 haplotype combination of haplotype J and the K2K2 haplotype combination of haplotype K were significantly better than those of other haplotype combinations. Haplotype N (N2N2), haplotype O (O2O2), haplotype P (P2P2), haplotype Q (Q2Q1), haplotype R (R1R1), haplotype S (S4S1), and haplotype T (T1T1) exhibited significantly better performance than other haplotype combinations in CD. The CW of haplotypes U2U1 and V1V2 was significantly better than that of other haplotype combinations. The TC of the W1W1 haplotype W and X1X1haplotype X haplotype combinations was significantly better than that of other haplotype combinations. The BW of the haplotype combination Y1Y1 of haplotype Y, Z1Z1 of haplotype Z, and AB3AB2 of haplotype AB were significantly higher than those of other haplotype combinations (*p* < 0.05), and there were no significant differences observed among the haplotype combinations of other haplotypes (*p* > 0.05). By comparing their positions, the functional annotation genes *PBX1*, *GABGR1*, *AADAT*, *TRNAS-GGA-82*, *KIAA1109*, *IGFBP3*, *REV3L*, *ARMC2*, *BAG3*, *TRNAG-UCC-59*, *TGFB2*, *TRNAG-UCC-34*, *KCNK9*, *FASTKD2*, *HACD3*, *MEGF11*, and *SLC8A1* were found to be located near significant haplotypes.

## 4. Discussion

Growth traits serve as key indicators in the trade and the breeding objectives of goats, influenced by numerous micropotent polygenes. Therefore, it is crucial to investigate the molecular markers and candidate genes associated with growth traits to enhance the genetic breeding of cashmere goats and support related industries [33,34]. They are affected by micropotent polygenes, and given that these traits are affected by micropotent polygenes, exploring molecular markers and candidate genes related to growth traits is of great significance for the genetic breeding of cashmere goats and related industries. Since haplotypes contain more LD information, it is more conducive to find variation loci associated with diseases or important economic traits in association analyses [18,19,20]. Current studies on cashmere goats are mainly focused on cashmere [14] and horn traits [35,36,37], but studies on growth traits are relatively scarce. To date, genome-wide association studies (GWAS) and polymorphism verification of candidate genes have been conducted on the growth traits of sheep. Significant single nucleotide polymorphisms (SNPs) and candidate genes, including *CAMK-MT*, *IGF-1*, *GH*, *GHR*, and *OSMR* [7,38,39,40,41], have been identified as significantly associated with morphological characteristics. Additionally, candidate genes such as *KITLG*, *CADM*, *MCTP1*, and *COL4A6* [34], have been implicated in the regulation of body height in Hu sheep. Furthermore, the candidate genes *MSRA*, *IQCH*, *TEK*, *LINGO2*, *PCDH10*, and *LGALSL*, among others, have shown significant associations with the morphological characteristics of Tibetan sheep and wild Argali [42].

Non-genetic factors, population stratification, kinship, and phenotypic records significantly influence the accuracy of genome-wide association analysis results [43]. In this experiment, SAS software was used to assess the effects of non-genetic factors (age, sex, herd, measured year) on growth traits (*p* < 0.05). However, since the herd was fed in groups based on sex, we included the group effect in the model to prevent over-correction. Additionally, data on BH, BL, CC, CD, CC, and TC were collected in 2023, and it was recorded only once; therefore, the year effect was not considered. Based on the ANOVA results, age and group were incorporated into the mixed linear model as fixed effects for correction to reduce false positives and improve the accuracy of the results. The inconsistency of the genetic background of study populations in GWAS will lead to population stratification, but population stratification is inevitable. Therefore, in this study, population stratification and inter-individual affinity coefficients were added to the mixed linear model as covariables and random environmental effects, respectively. Finally, the reliability of QQ-plot results was judged according to the degree of fitting between the expected value of the QQ-plot and the observed value and λ value [16]. Upon evaluating the aforementioned influencing factors, the anticipated value of the GWAS results aligned with the observed value. This alignment indicates that the model is robust and that these loci exhibit a significant correlation with growth traits.

In this study, genome-wide association analysis and haplotype analysis were performed on the growth traits of cashmere goats using resequencing data. A total of 284 SNPs significantly related to growth traits were detected. Interestingly, SNP chr7_29685358 was found to be linked to both chest circumference and chest width traits across the whole genome, and there was a positive correlation between chest circumference and chest width traits. The results are consistent with previous analyses of genome-wide association studies (GWAS) and *PRDM6* gene polymorphisms concerning growth traits in the Chinese Holstein cattle population, Karachai Goat, and IMCGs [44,45]. Through correlation analysis, this study found that there was a positive correlation between morphological characteristics and body weight traits. Still, no SNPs significantly correlated with both morphological characteristics and body weight were found in the GWAS results, which may be due to the small sample size for the GWAS analysis of the morphological characteristic traits. The accuracy of the analysis results could be improved by increasing the sample size. Haplotype analysis showed significant phenotypic differences in growth traits among the 24 haplotype combinations. However, the dominant haplotype combinations B2B1, G3G2, Q2Q1, S4S1, U2U1, V1V2, and AB3AB2 were heterozygous and could not be stably inherited from offspring, and so they were not considered molecular markers related to growth traits. The investigators used SNP-GWAS and haplotype GWAS to screen three key genes: *DHCR24*, *PLCB1*, and *SPATA9*. They also identified the hematopoiesis-related gene FLI1 through haplotype GWAS [46]. This indicates that overlapping markers from both methods enhance result reliability, while non-overlapping markers may reveal new, valuable associations. This study identified the genes *TGFB2*, *BAG3*, *ZEB2*, *KCNJ12*, *MIF*, *MAP2K3*, *HACD3*, and *MEGF11* as being associated with growth traits in IMCGs (Erlangshan type), influencing skeletal muscle growth and development.

Transforming growth factor-β is a superfamily that regulates cell growth and differentiation [47,48,49,50], and its biological functions in inflammation [51], embryonic development [52], and tissue repair have been extensively studied. The *TGFB2* gene, as a subunit of transforming growth factor-β, has a certain influence on growth and development. The results of the GWAS and quantitative PCR showed that *TGFB2* gene polymorphism was correlated with tibia length, bone mineral content, bone mineral density, and other traits of chickens, and the expression of the *TGFB2* gene was different in different breeds, tissues, and growth stages. The regulation of *TGFB2* gene expression was at its peak in the embryonic stage of the leg muscle tissue; it plays an important role in the proliferation of chicken leg muscle myoblasts [53,54]. Tang sequenced the promoter and full-length exon regions of *TGFB2* in chickens and discovered that both mutation sites within the promoter were significantly associated with body weight [55]. Furthermore, the expression of the *TGFB2* gene is dynamically regulated during muscle growth recovery and satellite cell differentiation, contributing to the regulation of bone and muscle growth [56,57]. Through transcriptome sequencing, bioinformatics analysis, and fluorescence quantitative PCR verification, *TGFB2* and *TGFB3* were found to be significantly correlated with the growth and development of southern yellow cattle, chickens [58], and the limb bone length of pigs [59]. Subsequently, through further gene editing experiments, it was found that after the gene was knocked out in mice, TGF-β signal transmission was interrupted, resulting in the loss of interphalangeal joints and skull hypoplasia of the mouse limbs, resulting in embryo death [60]. Knocking out *TGFB3* and *TGFB2* at the same time resulted in a reduction in the number of ribs, suggesting that these genes play an irreplaceable role in bone development.

Myocyte enhancer factor 2 (MEF2) contains four variable transcription factors (*MEF2A–D*) that cooperate with other transcription factors to control the development of skeletal muscle, cardiac muscle, and smooth muscle through protein interactions. It also acts as the end point of many intracellular growth factor regulatory signaling pathways to inhibit myoblast differentiation and promote proliferation in an antagonistic manner. Among these, *MEF2A* is a very important transcriptional regulatory factor that plays an important role in cell proliferation and differentiation, cell morphological changes, and other life processes [61,62]. It has been found that the *MEF2A* gene is significantly related to the body height traits of cattle, and its expression is different in the different growth stages and tissues of cattle [63]. In addition, another study found that in the context of the normal function of MEF2 members, the deletion, mutation, or knockout of *MEF2A* gene function still leads to a distinct disease phenotype, indicating that *MEF2A* has an irreplaceable function [64].

The zinc finger e-box-binding homeobox 2 (*ZEB2*) protein is a protein that belongs to the ZEB protein family and is involved in various metabolic regulation processes, including cell formation, growth, differentiation, and apoptosis [65,66]. This study, based on GWAS analysis, found that this gene was significantly correlated with the body weight traits of cashmere goats, and the results were consistent with the conclusion that the *ZEB2* gene affects the body weight traits of cattle and Hu sheep [67,68]. In a study on pig growth and development, this gene was associated with psoas muscle depth (*LMD*). Additionally, overexpression of *ZEB2* overexpression has been found to have a positive effect on the skeletal muscle differentiation of pluripotent stem cells and adult myogenic progenitors [69] and is involved in the regulation of human bone development [70].

Introverted rectified potassium channel family (*KCNJ*) genes play an important role in cell regulation, including cell volume, electrical excitability, and insulin secretion [71], and are generally expressed in animal cardiomyocytes and neuronal products. Based on SNP-GWAS and CNV-GWAS, it was revealed that *KCNJ12* is significantly associated with the shin and trunk traits of chickens and plays an important role in body height traits and muscle development in cattle [72,73]. Furthermore, this study found that *KCNJ12* is correlated with body weight traits in cashmere goats based on SNP-GWAS analysis. Notably, in the expression profiles of various bovine tissues and primary bovine skeletal muscle cells, *KCNJ12* is generally highly expressed in muscle cells. In primary bovine skeletal muscle cells, the expression of *KCNJ12* in the differentiation medium is progressively upregulated compared to the growth medium, indicating that the *KCNJ12* gene is involved in the differentiation of bovine muscle cells [73].

Weight gain is closely related to obesity, fat deposition, muscle development, and skeletal muscle growth. The candidate genes *MIF*, *MAP2K3*, *HACD3*, and *MEGF11* excavated in this study are related to obesity and fat deposition. For instance, the expression levels of these genes in freshly isolated adipocytes and pre-culture adipocytes are positively correlated with adipocyte diameter [74]. The results of the *3T3L1* lipid cell line showed that glucose could stimulate the expression of *MIF*. The Sakaue study found that haplotypes of both promoter polymorphisms of the *MIF* gene are associated with obesity, suggesting that increased expression of this gene may be the result of metabolic dysregulation in obesity. The process of fat formation is regulated by a variety of MAP kinase signaling pathways, and studies have shown that p38 MAP kinase can stimulate and inhibit fat formation [75]. Furthermore, Mitogen-Activated Protein (MAP) kinases serve as critical mediators in signal transduction pathways and are integral to the regulation of cellular processes such as growth, proliferation, differentiation, and apoptosis [76]. It was found that a polymorphism of snp (rs11652094) in the *MAP2K3* region is correlated with body weight in Caucasian people, and the expression level of *MAP2K3* in adipose tissue is positively correlated with body weight. In vitro studies of the cloned *MAP2K3* promoter have indicated that gene expression is upregulated during adipogenesis. In addition, it was found that the *HACD3* protein is involved in the production of ultra-long chain fatty acids with different chain lengths [77], and that *MEGF11* is related to feed conversion [78], which indirectly affects the body weight of domestic animals. Consequently, it can be inferred that the aforementioned genes possess the potential to function as molecular markers and candidate genes influencing the growth traits of cashmere goats.

## 5. Conclusions

In this study, a total of 284 SNPs; the haplotype combinations A2A2, C2C2, E2E2, F2F2, I2I2, J2J2, K2K2, N2N2, O2O2, P2P2, R1R1, T1T1, W1W1, X1X, Y1Y1, and Z1Z1; and the candidate genes *TGFB2*, *BAG3*, *ZEB2*, *KCNJ12*, *MIF*, *MAP2K3*, *HACD3*, and *MEGF11* were found to be significantly correlated with growth traits based on genome-wide association analysis and haplotype analysis. These molecular markers and candidate genes can serve as indicators associated with the growth traits of cashmere goats. Their application has the potential to enhance the growth characteristics of these animals, thereby increasing the precision of early selection processes and improving the economic efficiency of breeding programs.

## Figures and Tables

**Figure 1 vetsci-11-00428-f001:**
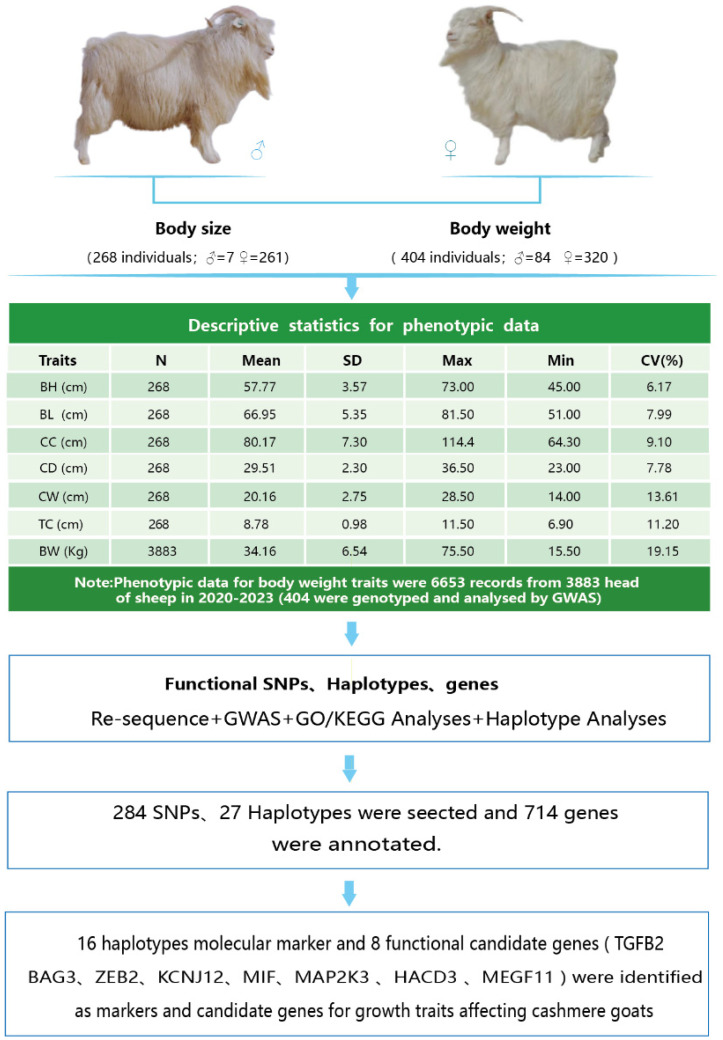
Analysis process. Body Height (BH), Body Length (BL), Chest Circumference (CC), Chest Depth (CD), Chest Width (CW), Tube Circumference (TC), Body Weight (BW). Considering that multiple body weight measurements were recorded for the same individual, in order to dissect their permanent environmental effects, their breeding values (breeding values + residuals) were derived using the repetitive force model for subsequent GWAS analyses.

**Figure 2 vetsci-11-00428-f002:**
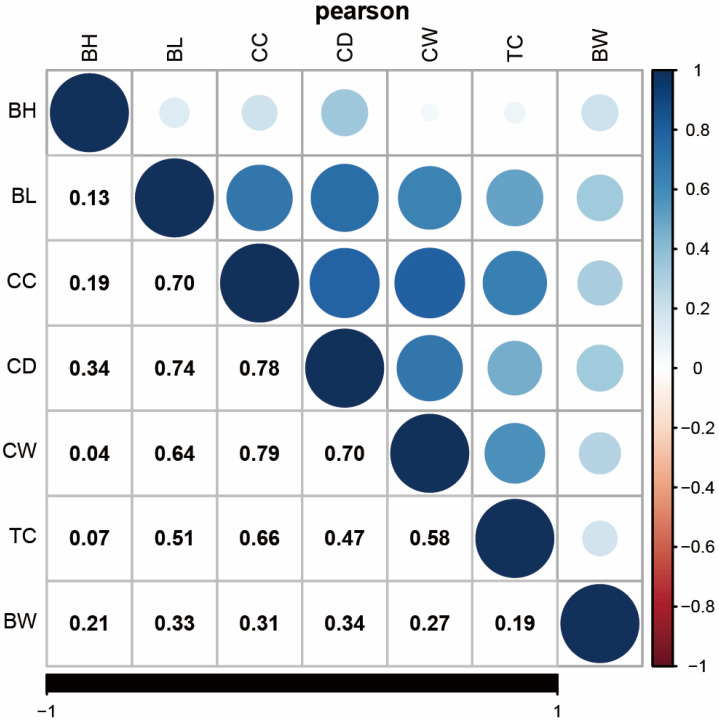
Correlation analysis of growth indexes of IMCGs.

**Figure 3 vetsci-11-00428-f003:**
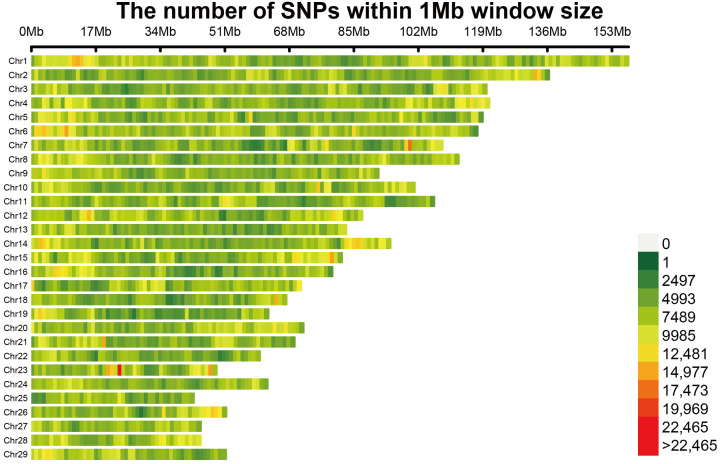
Distribution of SNPs in the 1 Mb window of chromosomes, with the left Y axis representing chromosome names and the upper X axis representing window sizes.

**Figure 4 vetsci-11-00428-f004:**
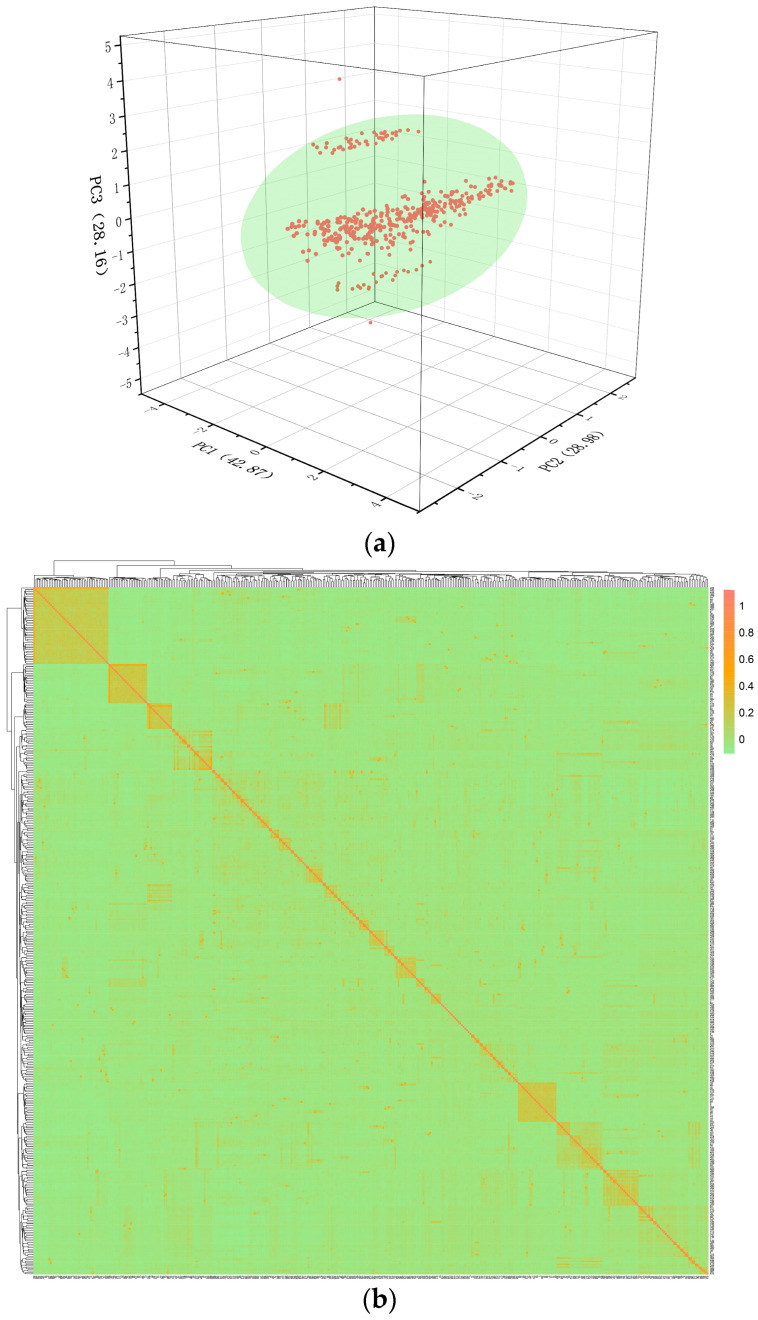
Population structure and relationship analysis of IMCGs (ErIangshan type). (**a**) Principal component analysis results diagram of IMCGs (ErIangshan type); (**b**) G matrix Heat map of IMCGs (ErIangshan type) in the conserved population. Each small square indicates the kinship value between different individuals. The closer the color of the square to red, the closer the kinship between individuals.

**Figure 5 vetsci-11-00428-f005:**
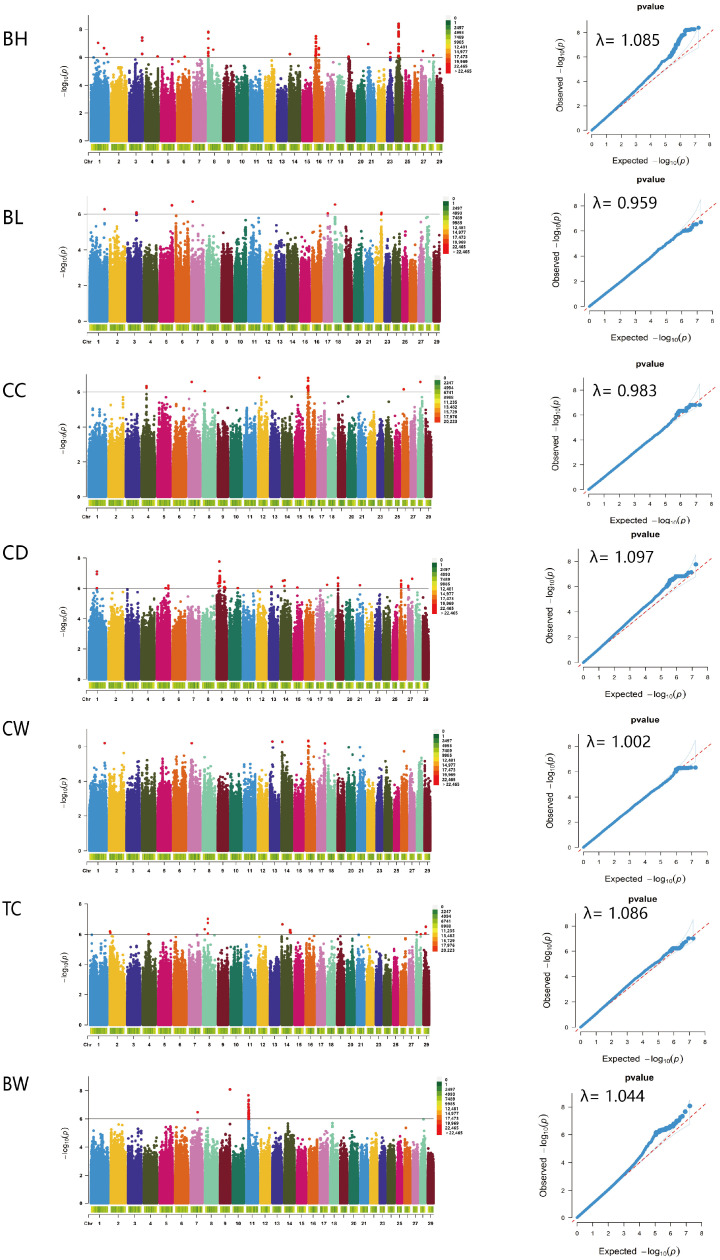
Manhattan plot and quantile-quantile (Q-Q) plot for growth traits. Body Height (BH), Body Length (BL), Chest Circumference (CC), Chest Depth (CD), Chest Width (CW), Tube Circumference (TC), Body Weight (BW). In the Manhattan plot (left), single nucleotide polymorphisms (SNPs) on different chromosomes are denoted by different colors (markers). Density is shown at the bottom of the Manhattan plot; the horizontal black line indicates a significant genome-wide association threshold (*p* = 1.0 × 10^−6^). Q-Q plots are displayed as scatter plots of observed and expected log *p*-values (right).

**Figure 6 vetsci-11-00428-f006:**
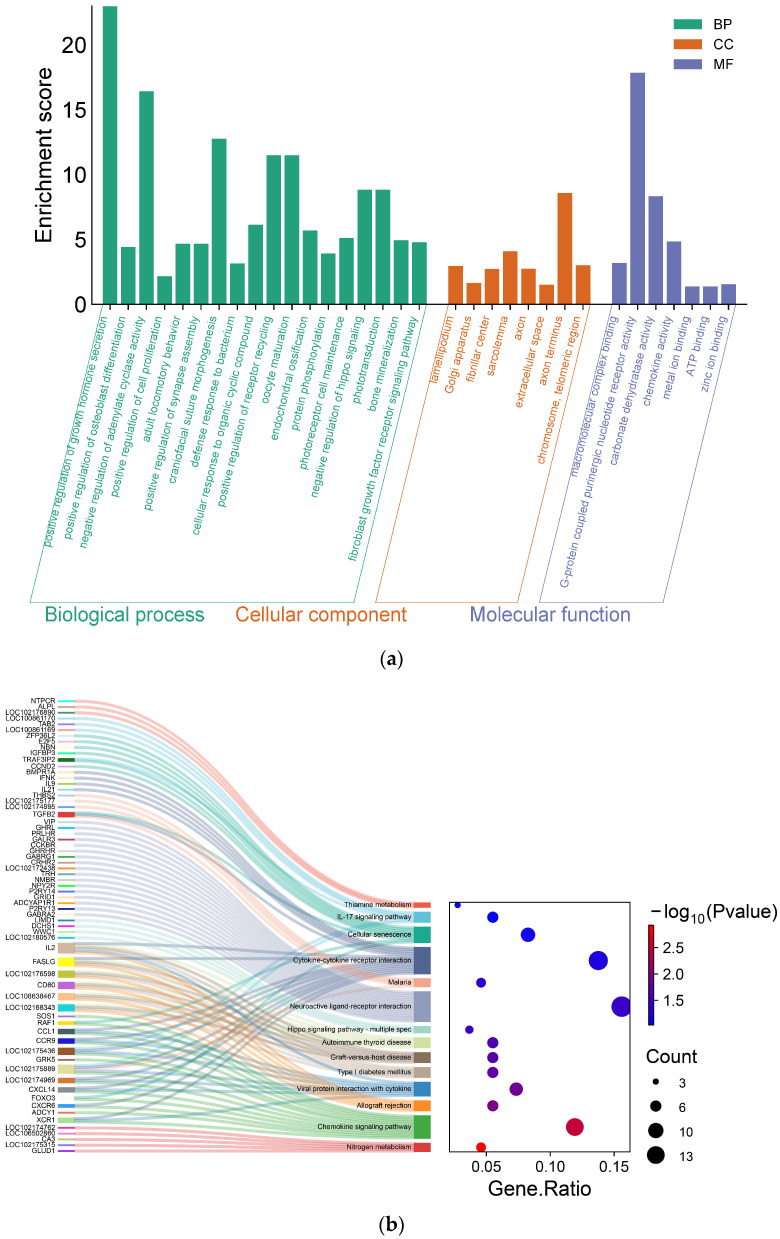
Enrichment analysis of growth traits of IMCGs (Erlangshan type). (**a**) Secondary classification histogram of Gene ontology (GO) enrichment analysis of candidate genes. (**b**) KEGG enrichment analysis diagram.

**Figure 7 vetsci-11-00428-f007:**
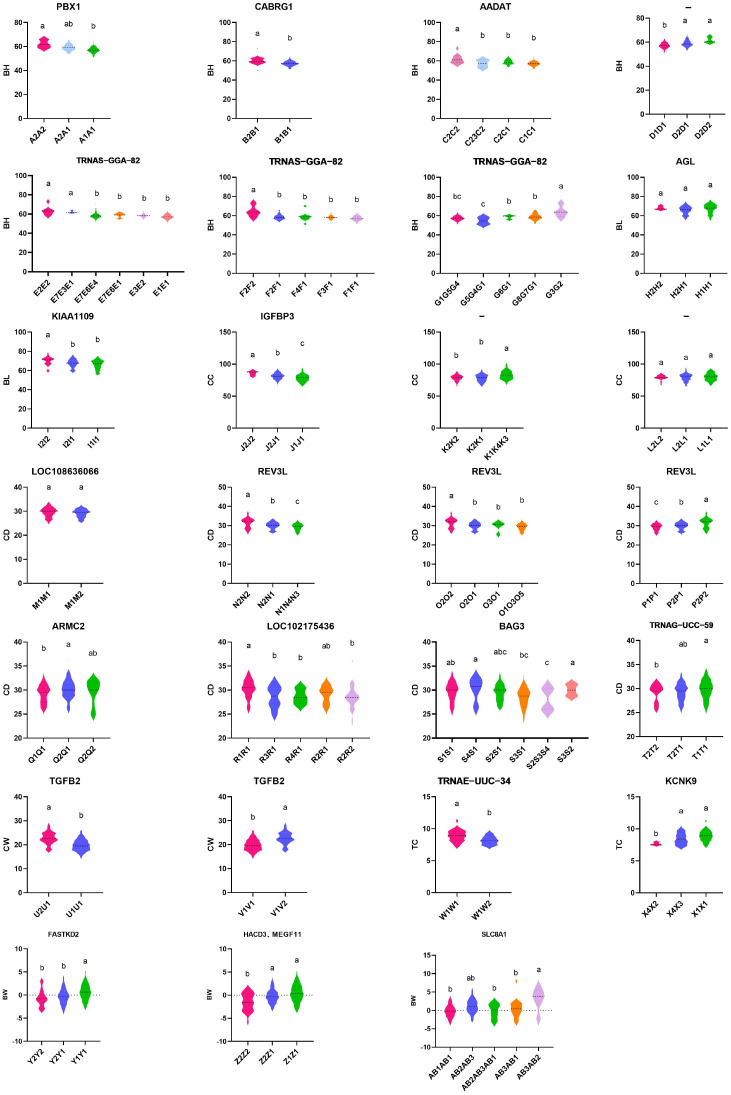
Association analysis of haplotype combinations with growth traits in IMCGs (Erlangshan type) (**a**–**c**); significant differences between genotypes indicated with different lowercase letters (*p* < 0.05). *x*-axis indicates haplotype combinations, *y*-axis indicates phenotypes corresponding to growth traits, and markers at the top of the graph are annotated candidate genes, which include *PBX1*, *GABGR1*, *AADAT*, *TRNAS-GGA-82*, *KIAA1109*, *IGFBP3*, *REV3L*, *ARMC2*, *BAG3*, *TRNAG-UCC-59*, *TGFB2*, *TRNAG-UCC-34*, *KCNK9*, *FASTKD2*, *HACD3*, *MEGF11*, and *SLC8A1*. Body Height (BH), Body Length (BL), Chest Circumference (CC), Chest Depth (CD), Chest Width (CW), Tube Circumference (TC), Body Weight (BW).

**Table 1 vetsci-11-00428-t001:** SNP loci and candidate genes significantly associated with the growth traits of IMCGs (Erlangshan type).

Traits	SNP	Chr	Position	Base Mutation	BETA	*p* Value	r2 (%)	Distance (bp)	Gene
BH	chr14_5226699	14	5,226,699	A > G	−1.831	5.98 × 10^−7^	2.526	−56,540	E2F5
	chr21_6110120	21	6,110,120	C > A	1.754	1.13 × 10^−7^	2.769	within	MEF2A
CC	chr4_44148386	4	44,148,386	G > A	2.274	4.65 × 10^−7^	2.513	−98,274	IGFBP3
	chr4_44149271	4	44,149,271	A > C	2.274	4.65 × 10^−7^	2.513	−84,250	IGFBP3
	chr4_44149291	4	44,149,291	G > A	2.274	4.65 × 10^−7^	2.513	−84,270	IGFBP3
	chr4_44149517	4	44,149,517	G > A	2.274	4.65 × 10^−7^	2.513	−84,496	IGFBP3
	chr4_44150099	4	44,150,099	C > T	2.274	4.65 × 10^−7^	2.513	−85,078	IGFBP3
	chr4_44151090	4	44,151,090	G > A	2.264	5.76 × 10^−7^	2.472	−86,069	IGFBP3
	chr7_29685358	7	29,685,358	A→C	−1.907	2.66 × 10^−7^	2.686	within	MSH3
	chr26_12222407	26	12,222,407	G→A	3.430	7.11 × 10^−7^	2.441	within	BAG3
	chr28_20591571	28	20,591,571	A→G	3.119	2.66 × 10^−7^	2.626	within	CCAR1
CD	chr26_12637966	26	12,637,966	G→T	−1.100	3.10 × 10^−7^	2.683	−7966	BAG3
	chr26_12637982	26	12,637,982	G→A	−1.100	3.10 × 10^−7^	2.683	408,077	BAG3
	chr26_12640600	26	12,640,600	C→T	−0.664	8.41 × 10^−7^	2.518	410,695	BAG3
CW	chr7_29685358A	7	29,685,358	A > C	−0.897	6.46 × 10^−7^	2.547	within	MSH3
	chr16_20378402	16	20,378,402	C > T	1.996	5.03 × 10^−7^	2.494	within	TGFB2
	chr16_20379075	16	20,379,075	A > G	1.996	5.03 × 10^−7^	2.494	within	TGFB2
	chr16_20384805	16	20,384,805	C > G	1.996	5.03 × 10^−7^	2.494	within	TGFB2
	chr16_20385646	16	20,385,646	C > T	1.996	5.03 × 10^−7^	2.494	within	TGFB2
	chr16_20388796	16	20,388,796	C > T	1.996	5.03 × 10^−7^	2.494	within	TGFB2
	chr16_20391015	16	20,391,015	C > T	1.996	5.03 × 10^−7^	2.494	within	TGFB2
	chr16_20392873	16	20,392,873	A > G	1.996	5.03 × 10^−7^	2.494	within	TGFB2
	chr16_20392878	16	20,392,878	C > T	1.996	5.03 × 10^−7^	2.494	within	TGFB2
	chr16_20395972	16	20,395,972	A > G	1.974	4.60 × 10^−7^	2.510	within	TGFB2
	chr16_20397072T	16	20,397,072	T > C	1.974	4.60 × 10^−7^	2.510	within	TGFB2
	chr16_20399832	16	20,399,832	G > A	1.996	5.03 × 10^−7^	2.494	within	TGFB2
	chr16_20400420	16	20,400,420	G > A	1.996	5.03 × 10^−7^	2.494	within	TGFB2
	chr16_20533282	16	20,533,282	G > A	2.059	9.36 × 10^−7^	2.378	−93,711	TGFB2
BW	chr2_84197799	2	84,197,799	G→A	1.689	9.68 × 10^−7^	2.366	21,935	ZEB2
	chr2_84199224	2	84,199,224	A→C	1.689	9.68 × 10^−7^	2.366	−23,360	ZEB2
	chr2_84205973	2	84,205,973	G→A	1.686	6.73 × 10^−7^	2.433	−30,109	ZEB2
	chr2_84220465	2	84,220,465	G→A	1.686	6.73 × 10^−7^	2.433	−44,601	ZEB2
	chr2_84226486	2	84,226,486	G→A	1.686	6.73 × 10^−7^	2.433	−50,622	ZEB2
	chr2_84230899	2	84,230,899	C→T	1.686	6.73 × 10^−7^	2.433	−55,035	ZEB2
	chr10_88442808	10	88,442,808	G→A	−0.999	6.22 × 10^−7^	2.547	355002, within	HACD3, MEGF11
	chr10_88442873	10	88,442,873	A→C	−0.984	9.09 × 10^−7^	2.472	354937, within	HACD3, MEGF11
	chr17_355945	17	355,945	G→A	2.036	1.82 × 10^−7^	2.686	within	MIF
	chr19_34727083	19	34,727,083	A→C	1.690	3.379 × 10^−8^	2.994	178913, 248058	MAP2K3, KCNJ12

Note: The r^2^ (%) value denotes the proportion of phenotypic variance attributed to the single nucleotide polymorphism (SNP). Positive values in the distance column signify the spatial separation between the SNP and the upstream gene, while negative values indicate the distance to the downstream gene. The table exclusively presents genes for which SNPs are located within the gene or are in closest proximity to the SNP, thereby identifying potential candidate genes, which are as follows: *E2F5*, *MEF2A*, *IGFBP3*, *MSH3*, *BAG3*, *CCAR1*, *BAG3*, *MSH3*, *TGFB2*, *ZEB2*, *HACD3*, *MEGF11*, *MIF*, *MAP2K3*, and *KCNJ12*.

## Data Availability

The supporting data of this study are available from the corresponding authors upon request.

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
