# Peer review of "Combined Genome-Wide Association Study and Haplotype Analysis Identifies Candidate Genes Affecting Growth Traits of Inner Mongolian Cashmere Goats"

_vetsci, 2024, doi:10.3390/vetsci11090428_

Round 1

Reviewer 1 Report

Comments and Suggestions for Authors

This research paper combined SNP-GWAS and haplotype analysis to reveal molecular markers and candidate genes affecting growth traits in cashmere goats.

The following are all my views and modifications on this work. 

1.The abbreviations of proper nouns are used in several places in the paper, including LD, HWE, Ho, and PIC, please add the full name.

2.The headings of 2.7 and 3.5, “Association analysis of haplotypes and growth traits”, are not properly formulated and need to be corrected.

3.Some gene names are not italicized in the text, such as those in 3.5, which needs to be revised.

4.In section 3.2, there is an error in the way the codes are described in parentheses.

5.The Discussion section did not mention the advantages of using both GWAS and haplotype analysis to identify molecular markers and candidate genes. These advantages need to be added to make the article more fluent and organized.

6.The English language must be thoroughly revised to ensure logical coherence. In other words, a good paper, but not prepared seriously.

Comments on the Quality of English Language

A good paper, but not prepared seriously.

Author Response

Thank you for your letter and for there rviewers’comments concerning our manuscript entitled “Combined Genome-wide association study and Haplotype analysis identifies candidate genes affecting growth traits of Inner Mongolia cashmere goats” (ID vetsci-3175067) .Those comments are all valuable and very helpful for revising and improving our paper, as well as the important guiding significance to our researches.We have studied comments carefully and have made correction which we hope meet with approval. Revised portion are marked in red in the paper. The main corrections in the paper and the responds to the editor and reviewer's comments are as flowing:

Comments 1: The abbreviations of proper nouns are used in several places in the paper, including LD, HWE, Ho, and PIC, please add the full name.

Response 1: We sincerely thank you for your careful reading. The content of "Abbreviations" is added after the content of Conclusion, and the full name of many proper nouns such as LD, HWE, Ho and PIC is supplemented in the content of" Abbreviations" on lines 504-522.

Comments 2: The headings of 2.7 and 3.5, “Association analysis of haplotypes and growth traits”, are not properly formulated and need to be corrected. 

Response 2: Thank you for the helpful comments and we agree with your point. Following your suggestion, the title "Association Analysis of Haplotype and Growth Traits" has been revised to "Association Analysis of Haplotype combination and growth Traits".

Comments 3: Some gene names are not italicized in the text, such as those in 3.5, which needs to be revised.

Response 3: All gene names in the text were modified to italics.

Comments 4: In section 3.2, there is an error in the way the codes are described in parentheses, please revise.

Response 4: It has been amended to (-- geno, 0.05 -- maf 0.05, --hwe 1e-6, -- mind 0.1).

Comments 5: The discussion part of the article did not mention the advantages of mining important molecular markers and candidate genes by using GWAS and haplotype analysis at the same time, which needs to be added to make the article more fluent and organized.

Response 5: Thank you for your advice and we are in agreement with you. This is discussed and added to lines 407-414 of the text, as follows: The investigators used SNP-GWAS and haplotype GWAS to screen three key genes: DHCR24, PLCB1, and SPATA9. They also identified the hematopoiesis-related gene FLI1 through haplotype GWAS [47]. This indicates that overlapping markers from both methods enhance result reliability, while non-overlapping markers may reveal new, valuable associations. This study found that GWAS and haplotype analysis identified the TGFB2, BAG3, ZEB2, KCNJ12, MIF, MAP2K3, HACD3, and MEGF11 genes as associated with growth traits in IMCGs (Erlangshan type), influencing skeletal muscle growth and development.

Comments 6: The English language must be thoroughly revised to ensure logical coherence. In other words, a good paper, but not prepared seriously.

Response 6: We feel great thanks for your professional review work on our article. The descriptions in the article have now been revised, focussing mainly on the Introduction and Discussion sections, and red-flagged.

Reviewer 2 Report

Comments and Suggestions for Authors

The study investigates the genetic basis of growth traits in Inner Mongolia Cashmere goats (Erlangshan type) through GWAS and haplotype analysis. Utilizing genome-wide resequencing data, the research identifies significant SNPs associated with growth traits. Principal component analysis and genetic relationship analysis reveal the population structure and relationships among the goats. The findings highlight several candidate genes that may influence growth, providing insights for breeding programs, nevertheless discussion should be improved. 

Some comments/suggestions were added to the pdf, below are the more significant:

Line 52-53

“which can reflect carcass quality, meat yield, and disease resistance [2,3].”

Perhaps also refer to feed efficiency?

Line 69-72

“Currently, a significant number of haplotype association analysis results have been discovered, and attributes linked to growth[21–25], immunity[26–28], reproduction[29], and diseases in domesticated animals and poultry have been identified to have haplotypes and candidate genes.”

Please rewrite the sentence.

Line 77-78

“excavated, laying a theoretical foundation for marker‐assisted selection and whole genome selection of growth traits of IMCGs.”

The results obtained from one IMCGS type may not be directly applicable to the other types. This sentence should be rewritten to reflect this possibility.

Line 219

The legends and titles of the axes of the figures are very difficult to visualise, please improve their legibility.

Line 249 and 260

Please add a legend for the growth traits. e.g.: BW - Body Weight;...

Line 259

“Table 1. SNP loci and candidate genes significantly associated with the growth traits of IMCGs.”

I suggest this tittle: “Table 1. SNP loci and candidate genes significantly associated with the growth traits of Erlangshan type IMCG.”

The study was only carried out on the one IMCG type, so this should be reflected in the title of the table. The results cannot be directly generalised to other IMCG types.

Line 263

Please add the name of the genes that appear in the table to the legend.

Line 309

Please add the name of the genes and of the growth traits to the legend of the figure.

Line 311

Growth traits are important economic traits that affect the productivity of livestock 311

and are one of the concerning traits of breeders[33,34].

Please review the sentence (three times the work “traits” in one sentence is too much).

Line 429

MAP kinase signaling pathways” is not only related to obesity and fat metabolism. MAP kinases are important mediators of signal transduction and play a key role in the regulation of many cellular processes, such as cell growth and proliferation, differentiation, and apoptosis (Kyosseva, 2004; doi: 10.1016/S0074-7742(04)59008-6). indeed, MAP kinase signaling network is integrated into most, if not all, defined homeostatic and regulatory responses of eukaryotic cells (Johnson, 2011; doi: 10.1021/cb100384z).

Discussion - Should be improved, because there are countless studies, some of which have already been published this year, which have found associations with growth. Genes such as myostatin, IFG1, GH and GHR, POU1f1, FASN, FGF7 (among other) are mentioned in the bibliography, not only in China but also in other countries. It might be interesting to investigate this a little further, not least because many of them are located in the somatotrophic axis and are related to some of the genes found in this study.

A similar study to the one in review was not cited and might be important to the discussion of the results.
-        Easa AA, Selionova M, Aibazov M, Mamontova T, Sermyagin A, Belous A, Abdelmanova A, Deniskova T, Zinovieva N. Identification of Genomic Regions and Candidate Genes Associated with Body Weight and Body Conformation Traits in Karachai Goats. Genes (Basel). 2022 Sep 30;13(10):1773. doi: 10.3390/genes13101773IF: 2.8 Q2 . PMID: 36292658; PMCID: PMC9601913.6       

Bibliography:

Several repeated references were found:

Reference

N1

N2

7. Zhang L.; Liu J.; Zhao F.; Ren H.; Xu L.; Lu J.; Zhang S.; Zhang X.; Wei C.; Lu G.; et al. GenomeWide Association Studies for Growth and Meat Production Traits in Sheep. PLoS ONE. 20138 e66569.

7

40

14. Wang F.H.; Zhang L.; Gong G.; Yan X.C.; Zhang L.T.; Zhang F.T.; Liu H.F.; Lv Q.; Wang Z.Y.; WangR.J.; et al. GenomeWide Association Study of Fleece Traits in Inner Mongolia Cashmere Goats. Anim Genet. 202152375–379.

14

36

34. Jiang J.; Cao Y.; Shan H.; Wu J.; Song X.; Jiang Y. The GWAS Analysis of Body Size and Population Verification of Related SNPs in Hu Sheep. Front. Genet. 202112 642552.

34

42

69. Cheng J.; Peng W.; Cao X.; Huang Y.; Lan X.; Lei C.; Chen H. Differential Expression of the KCNJ12 Gene and Association Analysis of Its Missense Mutation with Growth Traits in Chinese Cattle. Animals (Basel). 20199 273.

69

70

Please take time to review the formatting of the references carefully (remove extra spaces and apply the journal's rules).

Comments on the Quality of English Language

English is ok!

Author Response

I would like to thank the esteemed reviewers for their careful reading and constructive comments on our manuscript, ‘Combined Genome-wide association study and Haplotype analysis identifies candidate genes affecting growth traits of Inner Mongolia cashmere goats’ (ID vetsci-3175067). I have carefully considered these comments and revised the paper accordingly. Please review these materials and let me know if any further changes are required. Thank you once again for your valuable input and for the thorough review of my manuscript.

Comments 1: Line 52-53

“which can reflect carcass quality, meat yield, and disease resistance [2,3].”

Perhaps also refer to feed efficiency?

Response 1: Thank you for your professional review. According to the ideas you provided and literature review, it was found that growth traits can reflect the feed conversion rate of livestock, so this is added with the article on lines 53-55, as follows: Growth traits serve as critical economic indicators in livestock meat production, as they provide insights into carcass quality, meat yield, feed efficiency, and disease resistance.

Comments 2: “Currently, a significant number of haplotype association analysis results have been discovered, and attributes linked to growth[21–25], immunity[26–28], reproduction[29], and diseases in domesticated animals and poultry have been identified to have haplotypes and candidate genes.”

Please rewrite the sentence.

Response 2: The description of the passage is therefore ambiguous and the above has now been rewritten on lines 72-77, as follows: Utilizing haplotype analysis, numerous molecular markers and candidate genes related to growth, reproduction, immunity, and disease traits in livestock and poultry have been identified. Among these, MSTN, IGF1, BMP2, HHEX, FUBP3, and METTL3 have been associated with growth traits  [21–25]; IL0RB, IL23A, and PRNP have shown significant associations with immunity and disease traits [26–28]; and ARL5A and CACNB4 have been linked to reproductive traits [29].

Comments 3: Line 77-78

“excavated, laying a theoretical foundation for markerassisted selection and whole genome selection of growth traits of IMCGs.”

The results obtained from one IMCGS type may not be directly applicable to the other types. This sentence should be rewritten to reflect this possibility.

Response 3: Thank you for your helpful comments, and we agree with you. This study was conducted only in the Inner Mongolian velvet goat (Erwangshan-type) population, and therefore cannot be directly applied to other types, so this description was modified as follows on lines 82-84: These findings may provide a theoretical foundation for marker-assisted selection and whole genome selection of growth traits in IMCGs.

Comments 4: Line 219

The legends and titles of the axes of the figures are very difficult to visualise, please improve their legibility.

Response 4: Thank you for your valuable comments. In order to improve the legibility of the hiking press, we have enlarged and clarified the pictures in the article (Figure 4 b) and modified the figure notes as follows: Figure 4. Population structure and relationship analysis of IMCGs (ErIangshan type). (a) Principal component analysis results diagram of IMCGs (ErIangshan type); (b) G matrix Heat map of IMCGs (ErIangshan type) in the conserved population. Each small square indicates the kinship value between different individuals. The closer the colour of the square to red, the closer the kinship between individuals.

Comments 5: Line 249 and 260

Please add a legend for the growth traits. e.g.: BW - Body Weight;...

Response 5:Thank you for your valuable input. The figure notes in the article have been supplemented as follows: Figure 5. Manhattan plot and quantile-quantile (Q-Q) plot for growth traits. Body Height (BH), Body Length (BL), Chest Circumference(CC), Chest Depth (CD), Chest Width (CW), Tube Circumference (TC), Body Weight (BW). In the Manhattan plot (left), single nucleotide polymorphisms (SNPs) on different chromosomes were denoted by different colors (markers). density was shown at the bottom of the Manhattan plot; the horizontal black line presented a significant genome-wide association threshold (P = 1.0 x10-6). Q-Q plots were displayed as scatter plots of observed and expected log P-values (right).

Comments 6: Line 259

“Table 1. SNP loci and candidate genes significantly associated with the growth traits of IMCGs.”

I suggest this tittle: “Table 1. SNP loci and candidate genes significantly associated with the growth traits of Erlangshan type IMCG.”

The study was only carried out on the one IMCG type, so this should be reflected in the title of the table. The results cannot be directly generalised to other IMCG types.

Response 6: Thank you for your constructive suggestions, the title of Table 1 has been modified according to your suggestions as follows: SNP loci and candidate genes significantly associated with the growth traits of IMCGs (Erlangshan type).

Comments 7: Line 263

Please add the name of the genes that appear in the table to the legend.

Response 7: Thank you very much for your suggestion, and the gene names have now been added to the legend as follows: Note: The r² (%) value denotes the proportion of phenotypic variance attributed to the single nucleotide polymorphism (SNP). Positive values in the distance column signify the spatial separation between the SNP and the upstream gene, while negative values indicate the distance to the downstream gene. The table exclusively presents genes for which SNPs are located within the gene or are in closest proximity to the SNP, thereby identifying potential candidate genes, which are as follows: E2F5, MEF2A, IGFBP3, MSH3, BAG3, CCAR1, BAG3, MSH3, TGFB2, ZEB2, HACD3, MEGF11, MIF, MAP2K3 and KCNJ12.

Comments 8: Line 309

Please add the name of the genes and of the growth traits to the legend of the figure.

Response 8: Gene names have been added to the legend as follows: Figure 7. Association analysis of haplotype combinations with growth traits in IMCGs. abc Significant differences between genotypes with different lowercase letters (P<0.05). x-axis indicates haplotype combinations, y-axis indicates phenotypes corresponding to growth traits, and markers at the top of the graph are annotated candidate genes, which contain PBX1, GABGR1, AADAT, TRNAS-GGA-82, KIAA1109, IGFBP3, REV3L, ARMC2, BAG3, TRNAG-UCC-59, TGFB2, TRNAG-UCC-34, KCNK9, FASTKD2, HACD3, MEGF11, and SLC8A1. Body Height (BH), Body Length (BL), Chest Circumference (CC), Chest Depth (CD), Chest Width (CW), Tube Circumference (TC), Body Weight (BW).

Comments 9: Line 311

Growth traits are important economic traits that affect the productivity of livestock 311

and are one of the concerning traits of breeders[33,34].

Please review the sentence (three times the work “traits” in one sentence is too much).

Response 9: This description appears several times in the text and has been changed based on the suggestions you provided on lines 352-355, as follows:Growth traits serve as key indicators in the trade and breeding objectives of goats and are influenced by numerous micro potent polygenes. Consequently, it is of considerable importance to investigate molecular markers and candidate genes associated with growth traits to enhance the genetic breeding of cashmere goats and support related industries.

Comments 10: Line 429

①“MAP kinase signaling pathways” is not only related to obesity and fat metabolism. MAP kinases are important mediators of signal transduction and play a key role in the regulation of many cellular processes, such as cell growth and proliferation, differentiation, and apoptosis (Kyosseva, 2004; doi: 10.1016/S0074-7742(04)59008-6). indeed, MAP kinase signaling network is integrated into most, if not all, defined homeostatic and regulatory responses of eukaryotic cells (Johnson, 2011; doi: 10.1021/cb100384z).

Response 10: Thank you for your careful review and guidance, have now reviewed and gained much from the literature on MAP kinase signaling pathways that you recommended and have drawn on this literature in the article on lines 481-483, as follows: Furthermore, Mitogen-Activated Protein (MAP) kinases serve as critical mediators in signal transduction pathways and are integral to the regulation of cellular processes such as growth, proliferation, differentiation, and apoptosis [79].

②Discussion - Should be improved, because there are countless studies, some of which have already been published this year, which have found associations with growth. Genes such as myostatin, IFG1, GH and GHR, POU1f1, FASN, FGF7 (among other) are mentioned in the bibliography, not only in China but also in other countries. It might be interesting to investigate this a little further, not least because many of them are located in the somatotrophic axis and are related to some of the genes found in this study.

回复:Thank you for the relevant ideas and information provided, through the review of the literature found a new discovery and drew on the article on lines 363-371, as follows: To date, genome-wide association studies (GWAS) and polymorphism verification of candidate genes have been conducted on the growth traits of sheep. Significant single nucleotide polymorphisms (SNPs) and candidate genes, including CAMK-MT, IGF-1, GH, GHR, and OSMR [7,39–42], have been identified as being significantly associated with morphological characteristics. Additionally, candidate genes such as KITLG and CADM, as well as MCTP1 and COL4A6 [34], have been implicated in the regulation of body height in Husheep. Furthermore, candidate genes MSRA, IQCH, TEK, LINGO2, PCDH10, and LGALSL, among others, have shown significant associations with the morphological characteristics of Tibetan sheep and wild Argali [43].

③A similar study to the one in review was not cited and might be important to the discussion of the results.
- Easa AA, Selionova M, Aibazov M, Mamontova T, Sermyagin A, Belous A, Abdelmanova A, Deniskova T, Zinovieva N. Identification of Genomic Regions and Candidate Genes Associated with Body Weight and Body Conformation Traits in Karachai Goats. Genes (Basel). 2022 Sep 30;13(10):1773. doi: 10.3390/genes13101773IF: 2.8 Q2 . PMID: 36292658; PMCID: PMC9601913.6 

回复:Have read the literature you recommend and draw on it in the article on lines 392-397, as follows: Interestingly, SNP (chr7_29685358) was found to be associated with chest circumference and chest width traits across the whole genome, and there was a positive correlation between chest circumference and chest width traits. The results are the same as previously shown in the analysis of GWAS and PRDM6 gene polymorphisms and their association with growth traits in the Chinese Holstein cattle population、Karachai Goat and IMCGs [48,49].

④Several repeated references were found:

回复:Thank you for your careful reading of this manuscript and the corrections given, which have now been made to the duplicated literature.

Comments 11: Two places in the discussion section of the manuscript are for adding references, please add them.

Response 11: Thank you for your careful reading of this manuscript and for raising the issue of not adding references to the content on pages 431 and 454 in the article, the relevant literature has now been added as [59], [68, 69]. The literature on page 431 is cited in the Chinese master's thesis (Transcriptome analysis of growth and development of southern yellow cattle and their pituitary tissue, 2017, in Chinese), which is hereby borrowed within the article due to its reliable findings, but this reference was not added. In reviewing the literature, articles related to this article were found and borrowed [59].

Reviewer 3 Report

Comments and Suggestions for Authors

In the manuscript, genome-wide association analysis was performed on growth traits (body height, body length, chest circumference, chest depth, chest width, tube circumference, and body weight) of Inner Mongolia cashmere goats (Erlangshan type) based on resequencing data. Population genetic parameters were estimated, haplotypes were constructed for significant sites, and association analysis was conducted between haplotype and phenotype. But there are still some problems to be solved in this manuscript.

1. Please unify the writing of "Inner Mongolia cashmere goats" in the whole article. For example, in the "Simple Summary" section.

2. In the "Materials and Methods" section, you need to mark the version of R.

3. The description of the sequencing platform is incorrect. Please modify it.

4. In the 2.3 and 2.4 section, the "-6" in the p-value should be superscripted.

5. Whether the self-designed Excel program was used to calculate population genetic parameters, if yes, please indicate: Using the self-designed Excel program to calculate...

6. In the 2.7 section, you need to add a version of the LDBlockShow and SAS.

Comments on the Quality of English Language

none

Author Response

Thank you for your letter and for the comments concerning our manuscript entitled “Combined Genome-wide association study and Haplotype analysis identifies candidate genes affecting growth traits of Inner Mongolia cashmere goats”. (ID vetsci-3175067). Those comments are all valuable and very helpful for revising and improving our paper, as well as the important guiding significance to our researches.

Comments 1: Please unify the writing of "Inner Mongolia cashmere goats" in the whole article. For example, in the "Simple Summary" section.

Response 1: Thank you for your valuable comments, now unified in the full text of the ‘Inner Mongolia cashmere goats’ writing style.

Comments 2: In the "Materials and Methods" section, you need to mark the version of R.

Response 2: The version of R (V3.6.0) has been added.

Comments 3: The description of the sequencing platform is incorrect. Please modify it.

Response 3: Thank you for reading this article carefully and for pointing out any errors. The name ‘BGI MGI-T7’ has now been changed to ‘DNBSEQ-T7’.

Comments 4: In the 2.3 and 2.4 section, the "-6" in the p-value should be superscripted.

Response 4: Thank you for reading this article carefully. The -6 in the text has been superscripted.

Comments 5: Whether the self-designed Excel program was used to calculate population genetic parameters, if yes, please indicate: Using the self-designed Excel program to calculate...

Response 5: Thank you for your helpful comments, and we agree with you. Consequently, the text ‘the population genetic parameters were calculated by Excel software and tested to see whether they were in accordance with the Hardy-Weinberg equilibrium principle.’ has now been replaced by ‘the population genetic parameters were calculated by the self-designed Excel program and tested to see whether they were in accordance with the Hardy-Weinberg equilibrium principle’.

Comments 6: In the 2.7 section, you need to add a version of the LDBlockShow and SAS.

Response 6: The relevant content has now been added, as follows: LDBlockShow (V1.40)  SAS (V9.2).

Round 2

Reviewer 1 Report

Comments and Suggestions for Authors

The authors have adequately responded to all my comments.

Comments on the Quality of English Language

Minor editing of English language required.